# Mechanical Properties of GFRP Bolts and Its Application in Tunnel Face Reinforcement

**DOI:** 10.3390/ma16062193

**Published:** 2023-03-09

**Authors:** Huayun Li, Junfu Fu, Bingguang Chen, Xin Zhang, Zhiqiang Zhang, Lin Lang

**Affiliations:** 1School of Architecture and Civil Engineering, Xihua University, Chengdu 610039, China; 2School of Emergency Management, Xihua University, Chengdu 610039, China; 3Southwest Jiaotong University Chengdu Design Institute Co., Ltd., Chengdu 610039, China; 4Key Laboratory of Transportation Tunnel Engineering, Ministry of Education, Southwest Jiaotong University, Chengdu 610031, China

**Keywords:** tunnel in soft surrounding rocks, GFRP, tunnel face bolts, pull-out test, inverse analysis, pre-reinforcement parameters

## Abstract

As a new type of pre-reinforcement material for tunnel faces, glass fiber-reinforced polymer (GFRP) bolts can effectively and safely improve the stability of tunnel faces in soft surrounding rocks and speed up excavation. Therefore, in this paper, systematic research is carried out on the bond strength of GFRP bolts in tunnel faces and their relative pre-reinforcement parameters. Firstly, the effects of rebar diameter, anchorage length, and mortar strength on the bonding properties of GFRP bars were studied by indoor pull-out tests. The bond strength–slip curves under different working conditions were obtained, and the curves showed that the ultimate bond strength between GFRP bars and mortar was negatively correlated with the diameter of GFRP bars but positively correlated with the strength of the mortar. In addition, the increase in anchorage length led to a reduction in bonding strength. Secondly, inverse analysis was used to analyse the mechanical parameters of the bond performance of the anchor bars by the finite difference software FLAC3D, and the results indicated that 1/5 of the compressive strength of the GFRP bar grouting body can be taken as the ultimate bond strength to calculate the cohesive strength of the grout. Additionally, the formula of GFRP bar grouting body stiffness was revised. Finally, based on the results of laboratory tests and the inverse analysis, the numerical simulation analysis results showed that the optimal reinforcement configuration for a shallow buried tunnel face surrounded by weak rock is to use GFRP bars with a length of 17 m arranged in the center circle of the tunnel face with a reasonable reinforcement density of 1.0 bolt/m^2^. The calculation formula of the stiffness and cohesion strength of the GFRP bar grouting body and the reinforcement scheme proposed in this paper can provide a reference for the construction of shallowly buried rock tunnels in soft surrounding rock.

## 1. Introduction

Excavating a tunnel in soft surrounding rocks can easily loosen the core zone ahead of the tunnel face. The tunnel face would be deformed inward due to being subject to the pressure load arising from the overlaying soil and rocks and become unstable or even collapse. Therefore, it is critical to control the deformation of the surrounding soil and rocks during the excavation procedure. As verified by existing studies, the use of bolts as a reinforcement for the core soil and rocks could effectively control this deformation [1,2], which has given rise to the pre-reinforcement technology of tunnel face bolts.

In engineering projects, steel bars are not a preferred material for tunnel face bolts due to their high shear strength, resistance to cutting, and heavy weight. Conversely, GFRP is an ideal substitution for this purpose because of its light weight, high tensile strength, and easy-to-cut property. A number of researchers have carried out relevant studies on the anchorage performance and reinforcement parameters of GFRP bolts as a part of pre-reinforcement technology in tunnel engineering.

The bonding behavior is paramount to characterizing the mechanical property of GFRP bolts. Xue et al. [3] conducted a series of pull-out tests in different environments, indicating that the new type of GFRP bolts was slightly lower than steel bolts in bond behavior. Shi et al. [4,5] concluded that the proportion coefficient of bond strength between steel bolts and GFRP bolts is 1.2–1.5 by performing several pull-out tests in concrete specimens. The experimental tests by Zhang et al. [6] revealed the influence of bond length, bolt diameter, thread pitch, thread depth, sandblasting amount and concrete strength on the bond behavior of GFRP bolts and concrete. Nguyen et al. [7] indicated that the bond strength between a fiber and concrete matrix depends not only on the properties of the slurry but also on the shape and surface of the fiber. Bai et al. [8] carried out a full-scale pull-out test to investigate the bond and anchorage performance between GFRP anti-floating bolts and a concrete base plate. It was demonstrated that the working effect of GFRP bolts with concrete was superior to steel bolts. Shakiba et al. [9] found that the strength of concrete affects the failure modes during the pull-out test of the anchorage specimen. The failure of high-strength concrete is due to the shedding of the surface of GFRP bars, while the failure of low-strength concrete is due to the crushing of concrete. Kuang et al. [10] found that the ultimate bearing capacity and the average bond strength of the first interface of GFRP anchor bars are 25 kN, 9.4% higher than those of steel anchors, respectively, but the average bond strength of the second interface of steel anchors is slightly higher than that of GFRP anchor bars. Kim et al. [11] found that the appropriate water-to-cement ratio can improve the bond strength between GFRP bars and concrete. When the water-to-cement ratio is constant, the increase in concrete strength results in the increase in bond strength between concrete and GFRP bars. Ho et al. [12] adopted a numerical simulation method to analyze the influence of confining conditions on the mechanical properties and failure mechanism of bolt-mortar interfaces. Sjolander et al. [13] studied both the reinforcement and bond performance of GFRP bolts by varying the bending fracture energy and interactive fracture energy in the interfaces between shotcrete and the stratum. Yokota et al. [14] adopted laboratory shear tests and discontinuous deformation analysis to study the mechanical properties of the interface between rock bolts and the bond materials. By using the Kelvin solution, Wang et al. [15] proposed a new method to describe the mechanical property of the grouting GFRP bolts subject to the interaction of prestress and dynamic load.

In addition to the bonding behavior of GFRP bolts, a large number of studies have also focused on the application of pre-reinforcement technology in tunnel engineering. Chen et al. [16,17] studied the effect of tunnel face GFRP bolts on the stratum deformation caused by excavation by using a finite element simulation and found that the tunnel face GFRP bolts could effectively strengthen the core soil ahead of the tunnel face and thus control the stratum deformation caused by tunnel excavation. Chen et al. [18] studied the reinforcement effects of grouting, small pipes, front bolts, and other combination measures on the tunnel face. The results suggested that the front bolts outperformed the small pipes in the deformation control of the tunnel face. Li et al. [19,20] studied the influence of GFRP bolts’ reinforcement length, density, scope, and axial stiffness on the stability of a soft tunnel face based on theoretical analysis and numerical simulations. Zhang et al. [21] found that the safety factor of a tunnel face is positively correlated with the friction angle, cohesion, bolt length, and the number of bolts. However, if the bolt length and the bolt density exceed the optimal values, the safety level would not be greatly improved; on the contrary, it would increase the construction cost. Zhang et al. [22] proposed that bolt support can reduce both the support pressure of the tunnel face and the stress release degree of the surrounding rock and thus further reduce the failure zone range. Based on finite difference analysis, Kitchah et al. [23] found that the tunnel face bolts outperformed an umbrella arch in reducing the settlement of the tunnel crown, and using GFRP bolts to reinforce the advance core could reduce the deformation in the ground mass, including extrusion and settlement. Anagnostou et al. [24] compared both theoretical and numerical simulation results regarding the bond strength and proposed a design nomogram with the uniform arrangement of bolts in a homogeneous ground, constant or alterable bond strength, and various degrees of tunnel face stability in different erection sequences. Perazzelli et al. [25] suggested a new method to reinforce the tunnel face with bolts. This novel configuration method could be used for tunnel face bolts in any reinforcement spacing, length, and erection sequence in different environmental settings (e.g., rich water leakage environments, cohesive-frictional soils, soft surrounding rocks). Sun et al. [26] built up a mechanical mode of surrounding rocks for sea bottom tunnels by taking into account the combined influence of stratum reinforcement and leakage effects. The sensitivities of reinforcement parameters were studied on tunnel surrounding rocks. Xu et al. [27] proposed the scheme of adding prefabricated corrugated steel plates to the supporting structure. Prefabricated corrugated steel plates can effectively ensure the stability of the tunnel face and surrounding rock, reduce its convergence deformation, and ensure the safety of subway tunnel construction through numerical simulations and field verifications. Wang et al. [28] proposed a 3D rotational silo-torus stability model for the face analysis of circular tunnel faces constructed under cohesive-frictional soils. Centrifugal tests and numerical simulations showed that the model can accurately capture the failure modes of the roadway working face. Chen et al. [29] analyzed the potential influencing factors for the stability of tunnel faces based on the limit analysis technique. The analysis results indicated that the tunnel face bolts could significantly improve the stability of surrounding rocks.

The existing research mainly focuses on pull-out tests to study the bond behavior between steel/GFRP bolts and concrete. It should be noted that GFRP bolts and steel bolts greatly differ in bonding performance. Additionally, in tunnel construction, the cement mortar used in the bolt grout segments does not contain coarse aggregates. Considering the fact that the mechanical behavior of bolt grout (i.e., cement mortar) is generally different from that of concrete, the bonding performance between cement mortar and GFRP bolts should be further examined. Additionally, regarding the application of pre-reinforcement technology (e.g., the tunnel face GFRP bolts) in tunnel engineering, currently available studies mainly employed simulation-based methods, in which the bonding parameters are evaluated for the steel bolts instead of GFRP bolts. As mentioned previously, this may lead to significant discrepancies in the bond performance, entailing the need for a systematic investigation regarding the material parameters of GFRP bolts.

In this study, the effects of rebar diameter, anchorage length, and mortar strength on the bonding properties of GFRP bars were studied based on the pull-out tests of GFRP bars. Then, the bond pull-out tests of GFRP bars were simulated by the finite difference software FLAC3D 7.0. An inverse analysis was subsequently performed in order to reasonably quantify the mechanical parameters of GFPR bars with the mortar grouting body. Finally, the effects of length, reinforcement density, and range of GFRP reinforcement on the face of a shallowly buried tunnel in a soft stratum were studied by FLAC3D.

## 2. Experimental Program

### 2.1. Test Design

The literature [4] pointed out that the main factors influencing the bolt bond behavior include mortar strength and anchorage length. To study these factors, the test adopted GFRP bolts in diameters of 22 mm, 25 mm, and 28 mm in different lengths and cement mortar in strength grades of M15, M20, and M25 (M15 denotes that the mortar’s average comprehensive strength is 15 MPa). The test conditions are shown in Table 1.

### 2.2. Material and Specimen Parameters

#### 2.2.1. GFRP Parameters

Before the bond pull-out test on GFRP, 6 specimens of bolts in each diameter were taken for the reinforcement pull-out test to obtain the basic mechanical parameters of the reinforcement bolts, summarized in Table 2.

#### 2.2.2. Mortar Mix Proportions

All mortar samples in the test consisted of medium-grained sand, ordinary Portland cement, and polycarboxylic water reducer. For the mortar mix proportions in different strength grades, see Table 3.

#### 2.2.3. Specimen Parameters

The test shape of the specimen was cuboid, with four different sizes: 0.3 m × 0.3 m × 0.5 m, 0.3 m × 0.3 m × 1 m, 0.3 m × 0.3 m × 1.5 m, and 0.3 m × 0.3 m × 2 m. For each size, three samples were produced for each dimension following the ‘Standards for Test Method for Concrete Structures’ [30], as shown in Figure 1. In addition, the steel casing was set at the free end of the bar in order to avoid the bar being clipped during the loading process.

### 2.3. Loading and Measurement 

A hollow jack was used for one-way, stepwise loading until the failure of the reinforcement bolts, with each loading step at about 0.2 MPa. In the test process, the bolt pull-out displacement Δu at each loading step was measured and recorded, as illustrated in Figure 2.

## 3. Experimental Results and Discussion

The test results are shown in Table 4.

The bond strength is calculated according to the following equation [3]:(1)$$\tau =\frac{F}{\pi ld}$$
where *F* is the pull-out resistance, *l* is the anchorage length, and *d* is the bolt’s diameter.

### 3.1. Analysis of the Influence of the Bolt Diameters on GFRP Bond Behavior 

Specimens with anchorage lengths of 0.5 m were selected for analysis. The ultimate bond strength of bolts with different diameters is shown in Figure 3. The bond strengths of GFRP bolts with diameters of 22 mm, 25 mm, and 28 mm are 3.21 MPa, 2.80 MPa, and 2.69 MPa, respectively, when the mortar strength is M15. The bond strengths of GFRP bolts with diameters of 22 mm, 25 mm, and 28 mm are 5.19 MPa, 4.75 MPa, and 4.71 MPa, respectively, when the mortar strength is M20. Regarding the M28 mortar, the bond strength of GFRP bolts with diameters of 22 mm, 25 mm, and 28 mm are 6.28 MPa, 5.97 MPa, and 5.35 MPa, respectively. It can be seen from the results that with the increase in bolt diameter, the ultimate bond strength decreases. That is because the relative bond area between the bolt and mortar decreases (Equation (2)) when the bolt diameter increases, thus leading to a decrease in the bond strength.
(2)$$s_{r}=\frac{4\pi d}{\pi d^{2}}=\frac{4}{d}$$

### 3.2. Influence of Mortar Strength on GFRP Bond Behavior 

The test results of specimens with anchorage lengths of 0.5 m and GFRP bolt diameters of 22 mm are adopted for analysis. The bond strength-sliding curves in different mortar strengths are shown in Figure 4, which indicates that the ultimate bond strength increases along with the increase in the mortar strength upon failure. That is because, in the pull-out process, the bond strength is mainly supported by the chemical adhesive strength in the interface between the bolt and mortar. Once the mortar strength increases, the chemical adhesive strength between the bolt and mortar increases, and so does the bond strength.

### 3.3. Influence Analysis of Anchorage Length on GFRP Bond Behavior

Figure 5 shows the bond behavior of the M20 mortar. With the increase in anchorage length, the bond strength was reduced in a nonlinear manner. This is because the bond strength is not uniformly distributed along the steel bar shaft but concentrated in the front end [31,32]. When the anchorage length is relatively short, the bond strength tends to be uniformly distributed. With the increase in the length, the bond strength between the GFRP bolt and anchorage body becomes nonlinearly distributed, thus leading to the insufficiency of the bond strength.

### 3.4. Analysis of Failure Modes of Specimens 

There are three failure modes of specimens after loading: GFRP bolts pulling out, GFRP bolts breaking, and mortar splitting. The failure modes and their proportions are shown in Figure 6 and Figure 7.

In the process of pulling out, a relatively larger radial stress occurs on the ribbed GFRP bolts. If the specimen strength and mortar thickness are not adequate, mortar split failure can occur, and the bond behavior of GFRP bolts is yet to give a full play. Therefore, in real engineering, it is important to reasonably select the mortar strength and anchorage length to avoid split failure.

## 4. Parameter Analysis of the GFRP Bolt Bond Behavior Based on FLAC3D

In FLAC3D, it is common to set up grouting body parameters for cable elements so as to reflect the bond behavior between the steel bars and mortar. However, GFRP bolts and steel bars differ from each other essentially, and so do their bond behaviors. During the actual construction period, grade M20 mortar with an anchorage length of 0.5 m is often adopted for the grouting body of tunnel face bolts because of its safety and cost. Therefore, based on the results from the laboratory tests on specimens with anchorage lengths of 0.5 m, this paper adopts FLAC3D to simulate the bond behavior and conducts inverse analysis to propose the parameters of the GFRP bolt bond behavior that are suitable in FLAC3D.

### 4.1. Numerical Model for the Pull-Out Test

The numerical model for the pull-out test is shown in Figure 8. The mortar specimen was simulated using a solid element with dimensions of 0.3 m × 0.3 m × 0.6 m. The Mohr–Coulomb model was employed as the constitutive model, and the normal constraints were put at the free end of the reinforcement bolts. The GFRP bolt was simulated in a cable element with an anchorage length of 0.5 m, and the ideal elastic–plastic model was adopted.

As shown in Figure 8, one end of the GFRP bar was pulled out with speed V until the connection between the bolt and the mortar specimen was broken [33]. Monitoring points were set on the contact surface between GFRP bars and mortar specimens to record the displacement changes of anchors during the drawing process. Finally, the bond strength of the bolt was calculated by writing the fish function to obtain the bond strength versus the slip displacement curve.

In FLAC3D, the bond strength between reinforcement bolts and mortar is mainly related to the four parameters of grout perimeter, grout frictional angle, grout stiffness in unit length, and grout cohesion in unit length, among which the grout perimeter is ascertained by the actual borehole diameter, D. Here, the borehole diameter D is taken as 60 mm as per engineering practices, and as the literature [34] points out when the confining pressure on the specimen is 0, the influence of the grout frictional angle on the bolt pull-out behavior is negligible, so this case is not discussed here.

Therefore, the inverse analysis mainly focuses on the grout stiffness *k*_g_ in unit length and the grout cohesion *c*_g_ in unit length to ensure that the simulation results of the bond strength–slip relationship resulting from numerical calculations are as close as possible to those from the laboratory test.

Assuming that no failure occurs in the interface between the grout and surrounding rocks, and reinforcement bolts are bonded tightly in the mortar, the grout cohesion of ordinary steel bolts can be calculated as per the following equation [33]:(3)$$c_{g}=\pi d\tau_{\max}$$
where *d* is the reinforcement bolt diameter; $\tau_{\max}$ is the ultimate bond strength, which is usually obtained by a pull-out test and can also be taken as 0.5 times the grout’s compressive strength when there is a lack of pull-out test data.

For ordinary steel bolts, assuming that the slippage and the pull-out resistance are in a linear relationship, the grout stiffness can be inferred by elastic mechanics and calculated as follows [33]:(4)$$k_{g}=\frac{2\pi G}{\ln(1+2t/d)}$$
where *G* is the grout’s shear modulus, *d* is the reinforcement bolt’s diameter, and *t* is the grout’s thickness.

Meanwhile, in FLAC3D, due to the influence of the relative shear displacement between the host-domain grid points and the borehole surface, it is a common practice in the calculation of grout stiffness to multiply Formula (4) by the reduction factor λ (see Equation (5)). For ordinary steel bolts, the reduction factor λ is often taken as 0.1 [33].
(5)$$k_{g}=\frac{2\pi G}{\ln(1+2t/d)}\cdot \lambda$$

### 4.2. Inverse Analysis on cg, the Grout’s Cohesion in Unit Length

The grout cohesion is denoted as *c*_g1_ when $\tau_{\max}$ takes the ultimate value in the bond strength test and as *c*_g2_ when $\tau_{\max}$ takes the value of 0.5 times the mortar’s compressive strength. The values of grout cohesion under different conditions are summarized in Table 5, which indicates that *c*_g2_ is always greater than *c*_g1,_ and the average specific value is 2.4.

Based on the pull-out test results, the M20 mortar specimen was selected for this study. The grout stiffness remains unchanged in the process of the simulation. *c*_g1_ is multiplied by different reduction factors to study the influence of grout cohesion on the values of simulated extreme bond strength of reinforcement bolts. The simulation parameters are given in Table 6, and the simulation results are shown in Figure 9.

The inverse analysis indicated that the ultimate bond strength of the specimen increased with an increase in the grout cohesion. When the grout cohesion is taken as *c*_g1_, the ultimate bond strength corresponds with the test values. Additionally, from Figure 9, it can be seen that the ratio of different grout cohesions equals the ratio of the corresponding ultimate bond strength, i.e.,: (6)$$\frac{c_{gi}}{c_{gj}}=\frac{\tau_{i}}{\tau_{j}}$$

Equation (6) shows that if *c*_g2_ is taken as the grout cohesion, the ultimate bond strength of GFRP bolts in the numerical simulation is amplified by an average ratio of 2.4. Therefore, to ensure the simulation result is close to the real data, *c*_g2_ shall be divided by 2.4; in other words, in case of insufficiency of test data, the ultimate bond strength of reinforcement bolts shall be taken as 1/5 of the grout’s compressive strength.

### 4.3. Inverse Analysis of the Grout’s Stiffness

To obtain reasonable values of grout stiffness for GFRP bolts, the grout cohesion remained unchanged in the numerical simulation, and the bond strength–slippage curve in different grout stiffness was obtained by way of changing the reduction factors in Equation (5). The actual reduction factors can be determined by comparing them with the experimental data. The simulation results are shown in Figure 10. The results show that the analysis results are in good agreement with the test results when the reduction factor is within the range of $\left(\frac{1}{10},\frac{1}{15}\right]$

## 5. Numerical Analyses on Pre-Reinforcement Technology of Tunnel Face Bolts

Based on the research into the bonding behavior of GFRP bolts in the previous sections, a case study based on the Chenggong tunnel in Kunming, China, is investigated to study the systematic influence of the reinforcement parameters, including reinforcement length, reinforcement density and reinforcement range, on the structural behavior of a shallowly buried tunnel.

### 5.1. Establishment of Numerical Model

The numerical model is shown in Figure 11. Considering the boundary effects [35,36], the model is 74 m long, 50 m high and 60 m wide. The upper surface of the model is taken to the ground without any constraints. The four lateral faces are fixed by roller supports, while the bottom face is fixed by pinned supports. Except for the surrounding rock, which adopts the Mohr–Coulomb model as its constitutive mode, the other elements adopt the isotropic elastic model as their constitutive model. The parameters of the surrounding rock and initial support are shown in Table 7.

### 5.2. GFRP Bolts’ Reinforcement Length 

(1)Failure mode analysis of shallow tunnel face in soft surrounding rock

Figure 12a shows a failure model of a shallow tunnel face in soft surrounding rock established by Davis [37], where CD is the tunnel face, the right-angle trapezoid ABCE and the isosceles triangle ECD are the disturbed surrounding rock, *C* is the buried depth of the tunnel, *H* is the tunnel’s height, $\sigma_{t}$ is the minimum horizontal stress required for the stability of tunnel face, $\tan\alpha =\tan\beta =2\sqrt{(C+H)+1/4}$, and δ=π/2.

As the face bolts are only arranged in the range of the tunnel face, this paper simplifies the model in Figure 12a. The overlaying soil is simplified as a uniform load acting on the top of the tunnel face, and only the stability of surrounding rock within the height of the tunnel face is studied. The simplified model is shown in Figure 12b, and model parameters are the same as those in Figure 12a.

(2)Analysis of reinforcement length based on theoretical values

To make the reinforcement effective, the tunnel face bolts should pass through the fracture surface (line FED in Figure 12b) and enter a certain depth of stable surrounding rock. Based on the analysis, combined with the actual situation of the Chenggong Tunnel, the theoretical method is used to study the tunnel face bolts’ reinforcement length. The tunnel height *H* =12 m, and the depth to which the tunnel is buried is *C* =10 m. The maximum depth of the fracture surface is obtained as follows:(7)$$l_{CF}=\frac{H}{2\sin\alpha \cos\alpha }\approx 15.4\;m$$

It should be noted that there is no relevant specification to explain the length parameters of a bolt-reinforced tunnel face in tunnel engineering. Considering the similarity between the tunnel face support and the foundation pit support, the anchorage length of the bolt is determined to be not less than 1.5 m according to the ‘Technical Specification for Retaining and Protection of Building Foundation Excavations’ [38]. Therefore, the minimum theoretical reinforcement length in the tunnel vault is taken as follows:(8)$$l\geq l_{CF}+1.5=16.9\;m$$

Considering the safety of the tunnel, the face bolts’ length distributed along the tunnel height is uniformly taken as 17.0 m.

(3) Analysis of reinforcement length based on numerical simulation

The extrusion displacement of the target face while tunneling (y = 20 m) is shown in Figure 13. We can conclude that the farther the tunnel face is away from the target face, the smaller the extrusion displacement of the target face is. When the tunnel excavation is about 16 m away from the target face, the extrusion displacement of the target face is less than 1/10 of the maximum extrusion displacement. Therefore, it can be considered that when the tunnel face is 16 m away from the target face or more, the surrounding rock behind the target face can be considered undisturbed.

Figure 14 shows the distribution of the plastic zones of the surrounding rock after excavation without any supports. It indicates that the maximum distribution depth of the plastic zone in the tunneling direction is 17 m (the mesh size of the tunneling direction is 1 m per grid).

The reinforcement length of the tunnel face bolts should not only meet the specification requirements but also meet the requirements of the depth of the plastic zones and undisturbed surrounding rock. Therefore, based on the theoretical analysis and numerical calculation results, the reinforcement length of the tunnel face bolts is taken as 17 m.

### 5.3. GFRP Bolts’ Reinforcement Density

To prevent the extrusion failure of the tunnel face, the horizontal support stress provided by the tunnel face bolts should not be less than. Referring to existing studies [39,40,41], the number of tunnel face bolts required to maintain the stability of the tunnel face is calculated based on the minimum support stress of the tunnel face. Additionally, the reinforcement effectiveness under different face bolt numbers is discussed by numerical simulation to optimize the number of face bolts.

(1)Analysis of the face bolts’ number based on theoretical analysis

The calculation model of $\sigma_{t}$ is shown in Figure 11b and is calculated as follows [42,43]:(9)$$\sigma_{t}=\left(C+H/2\right)\gamma -2c_{u}\tan\alpha$$
where the tunnel depth *C* = 10 m, the tunnel face height *H* = 12 m, the volumetric weight γ=18 kN/m^3^, and the undrained shear strength $c_{u}=40$ kPa. Thus, $\sigma_{t}=121.47$ kPa.

The anchoring capacity of the tunnel face bolts is mainly dependent on the minimum value between the tensile strength of the reinforcement and the bond strength of the bolt–grout interface. Therefore, the number of tunnel face bolts shall be calculated according to Equation (10).
(10)$$\sigma_{t}=\min\left\{\frac{nA\sigma_{\mathrm{adm}}}{A_{s}};\frac{nS_{c}\tau_{\mathrm{adm}}}{A_{s}}\right\}$$
where $\sigma_{t}$ is the horizontal support stress of the tunnel face, *n* is the number of face bolts, *A* is the cross-area of a single GFRP bar, and $S_{c}$ is the lateral area of a single GFRP bar. Other calculation parameters are given in Table 8.

For a single bolt, the pull-out force and the shear force provided by the grout can be obtained by Equation (11):(11)$$\left\{\begin{array}{l} F_{l}=\sigma_{\mathrm{adm}}A=\sigma_{\mathrm{adm}}\times \frac{\pi d^{2}}{4}=226.94\;\mathrm{kN}\\ F_{s}=\tau_{\mathrm{adm}}S_{c}=\tau_{\mathrm{adm}}\times \pi dl=359\;\mathrm{kN} \end{array}\right.$$

Thereby, from Equation (12), we can obtain:(12)$$n=\frac{\sigma_{t}A_{s}}{\sigma_{\mathrm{adm}}A}=80$$

As the tunnel face bolt technology is a kind of temporary pre-reinforcement support, its factor of safety is taken as 1.4 based on the provisions of Technical Specification for Retaining and Protection of Building Foundations Excavations [37].
(13)n=80×1.4=112

(2)Analysis of face bolts’ number based on numerical simulation

Based on the theoretical number of tunnel face bolts, the effectiveness of reinforcement under five operating conditions, such as 77, 90, 112, 150, and 187 face bolts, is discussed to determine the optimized number of face bolts. The calculation model and bolt parameters are the same as above. Calculation results are shown in Figure 15 and Table 9.

The results indicate that the reinforcement effectiveness is proportional to the number of tunnel face bolts. When the number of bolts is less than 150, there is no obvious difference in reinforcement effectiveness under different working conditions. Additionally, when the bolt number reaches 150, the reinforcement effectiveness is significantly improved. However, when the bolt number continues to increase to 187, the reinforcement effectiveness does not improve significantly. Therefore, considering the safety and cost, the number of bolts should be 150; that is, the reinforcement density is 1.0 bolt/m^2^.

### 5.4. GFRP Bolts’ Reinforcement Range

As shown in Figure 16, in the full-face excavation process of the tunnel, the maximum extrusion deformation occurs in the lower middle part of the face. In engineering practice, full-section reinforcement is generally adopted to ensure construction safety. Although the reinforcement effectiveness is ideal, the cost-effectiveness is poor. Therefore, a reinforcement measure that can ensure construction safety and cost shall be determined.

Reinforcement effectiveness under three kinds of reinforcement range, i.e., upper reinforcement (Figure 17a), lower reinforcement (Figure 17b), and central round reinforcement (Figure 17c), were studied through numerical simulations.

The reinforcement density and reinforcement area in three working conditions are assumed to be equal, which are half of the tunnel face area and 1.0 bolt/m^2^, respectively. The calculation model and bolt parameters are the same as above.

Figure 18 shows reinforcement effectiveness under three working conditions, and the maximum extrusion displacements of the target face are also indicated in the figure. Both the upper reinforcement and the lower reinforcement can only control the extrusion displacement of the reinforced area, and the maximum extrusion displacement reduction rate is 14.0% and 40.8%, respectively; the reinforcements’ effectiveness on the unreinforced area is insignificant. The central round reinforcement has a good inhibition effect on the whole tunnel face, and the maximum extrusion displacement reduction rate is about 51.6%. In addition, the extrusion displacement curve form of the central round reinforcement is consistent with that of the full-face reinforcement. Considering the safety and cost, the central round reinforcement represents the optimized reinforcement range.

## 6. Conclusions

The effects of rebar diameter, anchorage length, and mortar strength on the bonding properties of GFRP bars were studied based on the indoor bond pull-out tests of GFRP bars. Then, the bond pull-out tests of GFRP bars were simulated by the finite difference software FLAC3D, and the inverse analysis of related anchorage parameters in numerical calculations was carried out. We also presented formulas for calculating the stiffness and cohesive strength of the GFRP bar grouting body. Based on the results of inversion analysis, the optimal reinforcement length, the reasonable reinforcement density, and the reasonable reinforcement range of the GFRP bar anchors were analytically determined in the shallow soft surrounding rock. The main conclusions include the following:(1)Laboratory test results show that when the anchorage length is constant, the ultimate bond strength between GFRP bolts and mortar is negatively correlated with bolt diameter and positively correlated with mortar strength, and when the bolt diameter and mortar strength are constant, the bond strength is in a nonlinear downtrend with the increase in the anchorage length.(2)Split failures occur in all M15 mortar specimens when the anchorage length is 0.5 m. As a result, that grade mortar shall be avoided as the grouting body during real construction. In addition, by analyzing the proportion of failure modes of specimens, it is concluded that in engineering practice, the strength of the mortar used as grouting material shall be greater than M20. When M20 mortar is used as grouting material, the minimum anchorage length of GFRP bolts with diameters of 22 mm, 25 mm, and 28 mm are 1.0 m, 1.5 m, and 2.0 m, respectively.Based on the ultimate bond strength of GFRP bars and mortar obtained from laboratory tests, the mechanical parameters of GFRP bars and the mortar grouting body are obtained by inverse analysis. Among them, the cohesive strength of the grouting body can be calculated by 1/5 of the compressive strength of the grouting body as the ultimate bond strength. The stiffness of the grouting body can be obtained by multiplying the stiffness formula of the grouting body of the reinforced bolt-concrete pull-out model with the reduction factor, and the range of the reduction factor is $\left(\frac{1}{10},\frac{1}{15}\right]$(4)A failure model of a shallow tunnel face in a soft stratum is established to determine the reasonable reinforcement length by simplifying the tunnel face failure model proposed by Davis. Considering the construction safety and cost, the optimum reinforcement length is taken as 17 m, the reasonable reinforcement density is 1.0 bolt/m^2^, and the GFRP bolts are anchored in the range of the central round.

## Figures and Tables

**Figure 1 materials-16-02193-f001:**
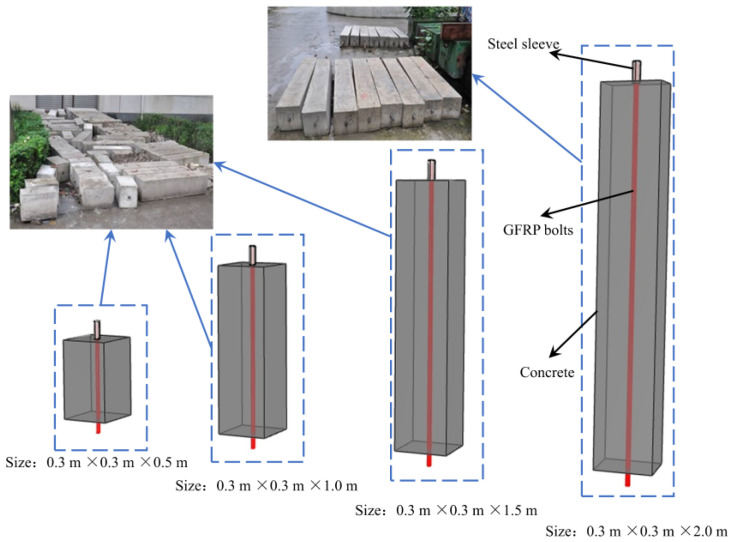
Specimen appearance.

**Figure 2 materials-16-02193-f002:**
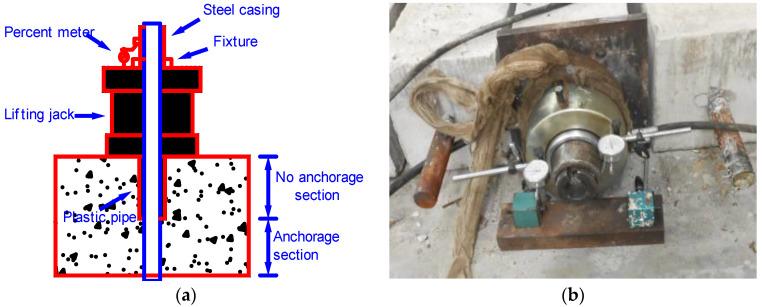
Hydraulic loading system. (**a**) Loading system diagram. (**b**) Specimen loading.

**Figure 3 materials-16-02193-f003:**
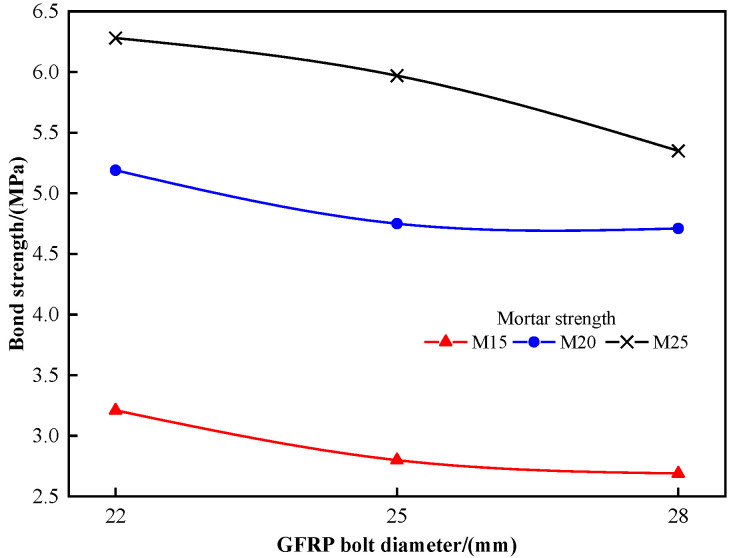
Relationship between bond strength and reinforcement diameter.

**Figure 4 materials-16-02193-f004:**
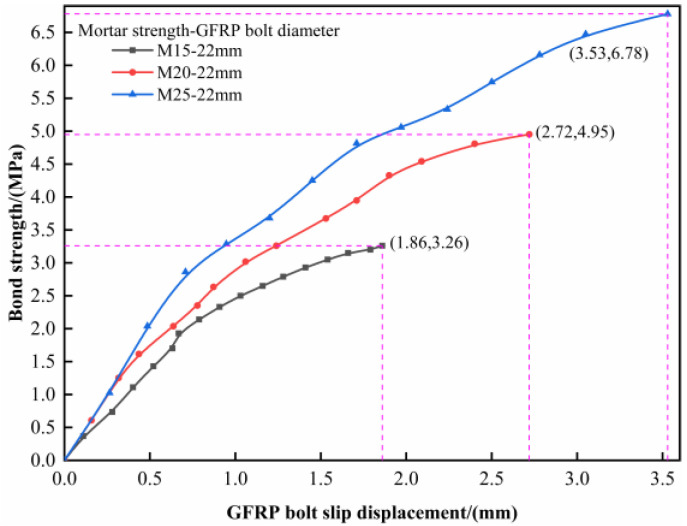
Relationship between bond strength and mortar strength.

**Figure 5 materials-16-02193-f005:**
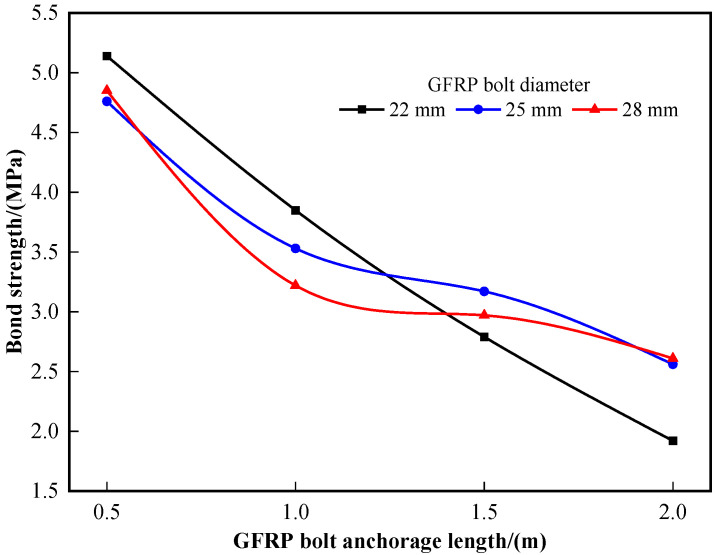
Relationship between anchorage length and bond strength.

**Figure 6 materials-16-02193-f006:**
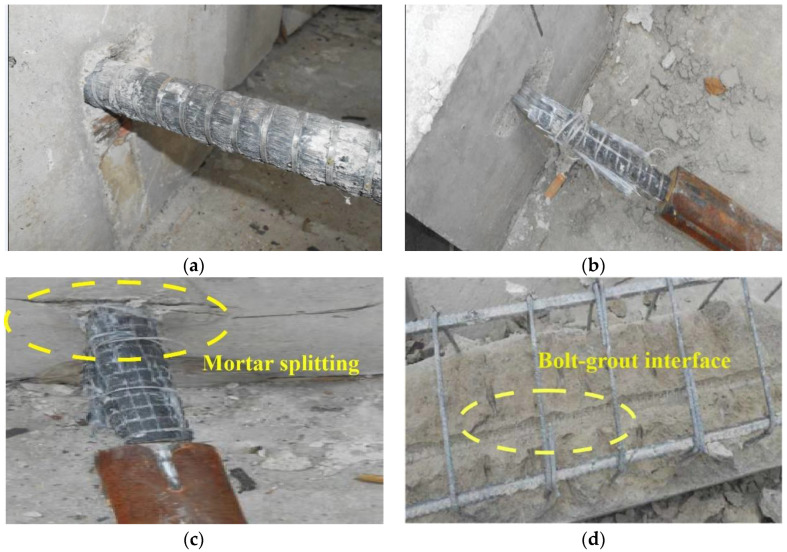
Failure modes of specimens. (**a**) GFRP bolt pulling out. (**b**) GFRP bolt breaking. (**c**) Mortar splitting. (**d**) Bolt-grout interface.

**Figure 7 materials-16-02193-f007:**
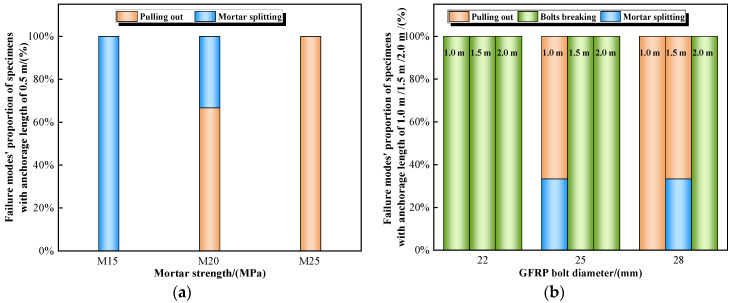
Proportion of specimens’ failure modes. (**a**) Anchorage length = 0.5 m. (**b**) Anchorage length = 1.0 m, 1.5 m, 2.0 m.

**Figure 8 materials-16-02193-f008:**
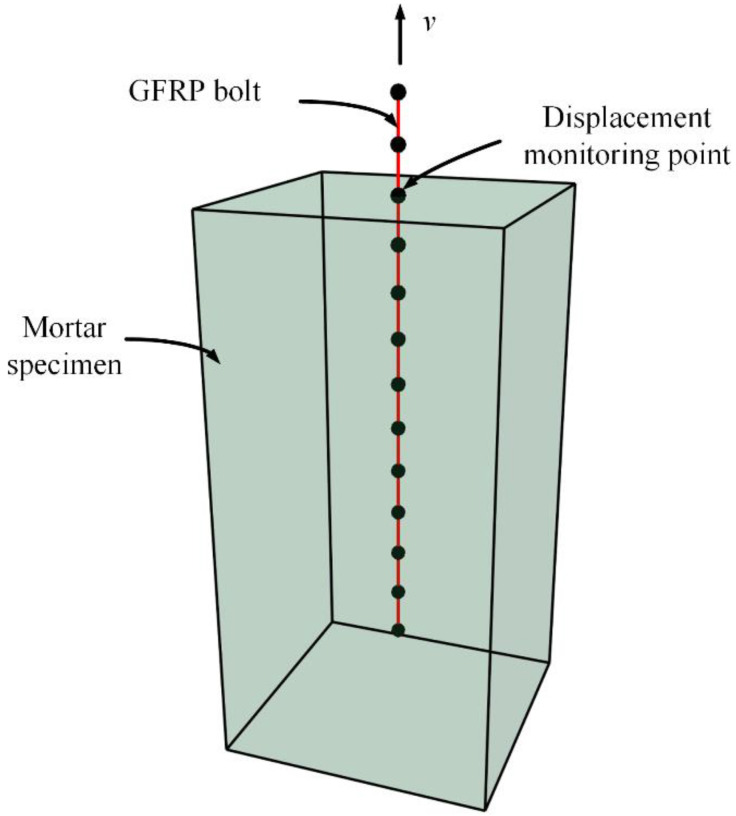
Numerical model of pull-out test on GFRP bolts.

**Figure 9 materials-16-02193-f009:**
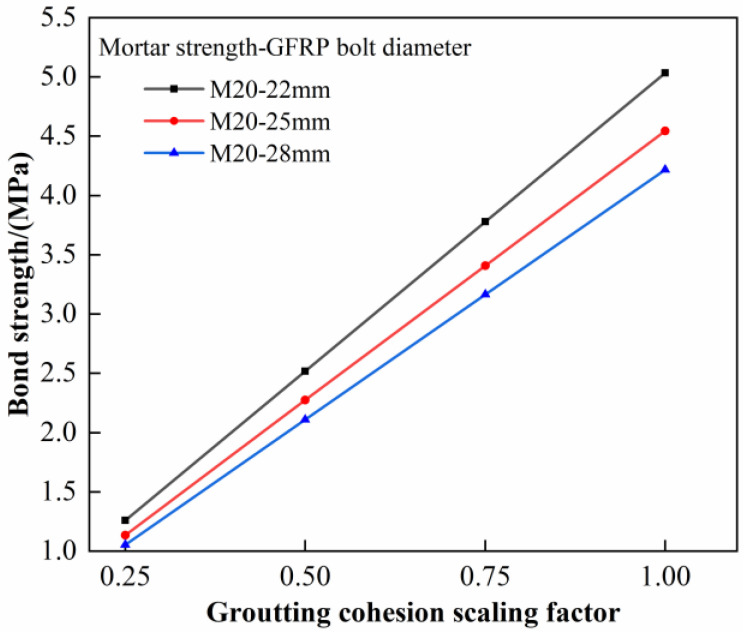
Relationship between computed ultimate bond strength and grout cohesion.

**Figure 10 materials-16-02193-f010:**
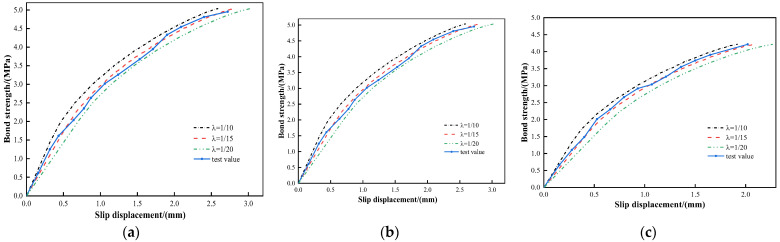
Inverse analysis results of the grout stiffness. (**a**) M20-22. (**b**) M20-25. (**c**) M20-28.

**Figure 11 materials-16-02193-f011:**
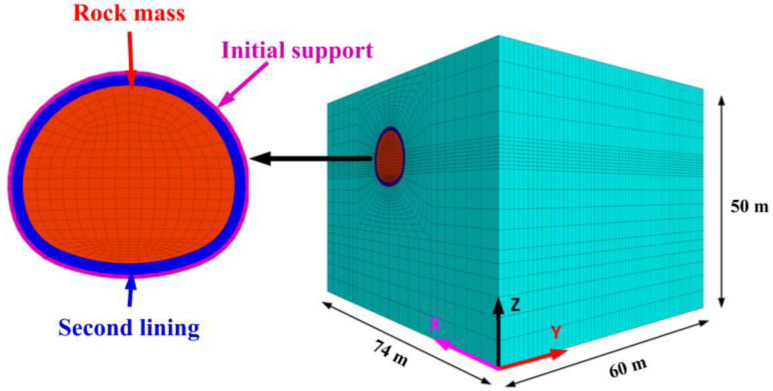
Numerical analysis of model.

**Figure 12 materials-16-02193-f012:**
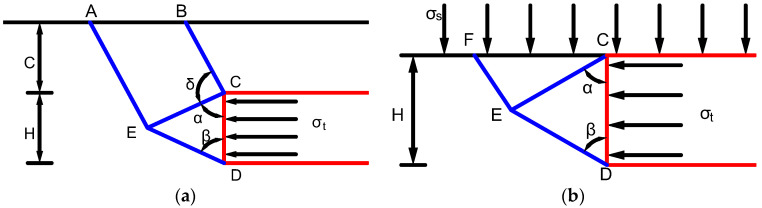
Failure model of shallow tunnel face in soft surrounding rock. (**a**) Failure mode diagram. (**b**) Simplified failure mode diagram.

**Figure 13 materials-16-02193-f013:**
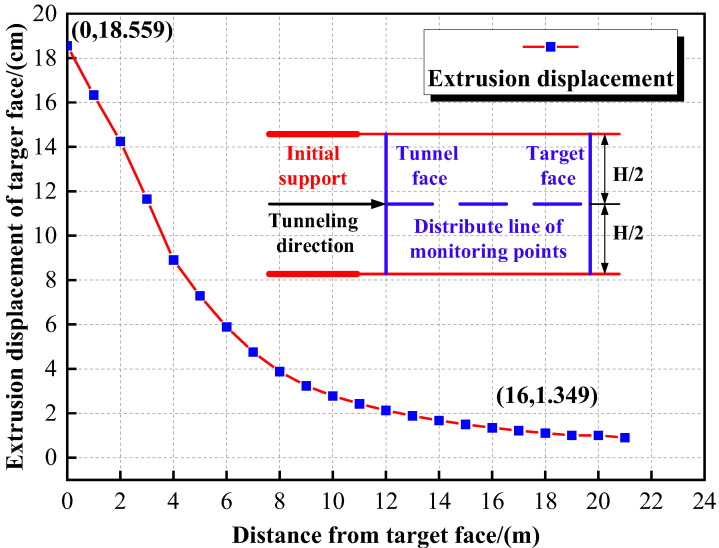
Extrusion displacement of the target face.

**Figure 14 materials-16-02193-f014:**
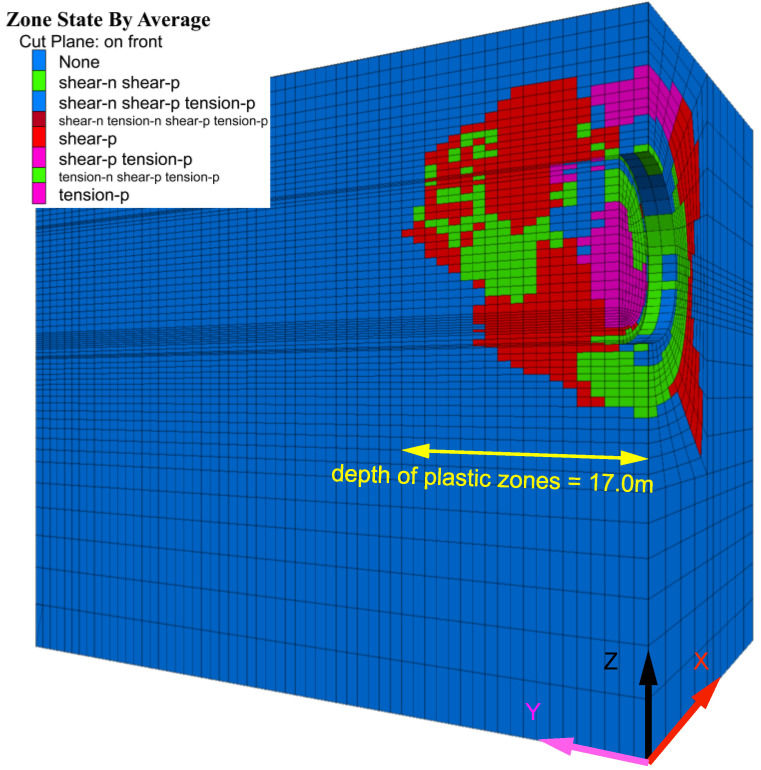
Distribution of plastic zones without support.

**Figure 15 materials-16-02193-f015:**
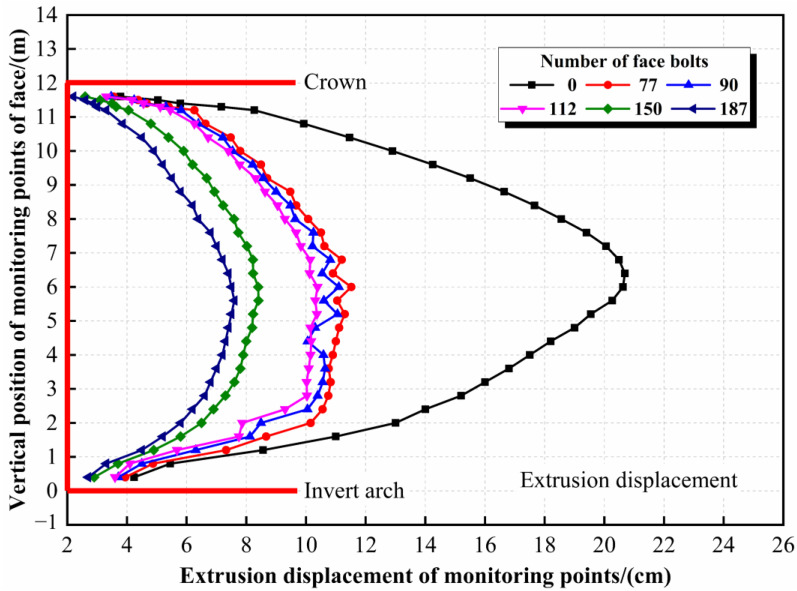
Extrusion displacement of target face under different conditions.

**Figure 16 materials-16-02193-f016:**
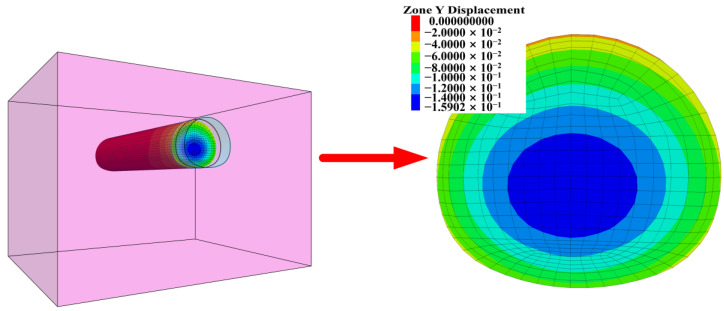
Extrusion displacement contour of tunnel face.

**Figure 17 materials-16-02193-f017:**
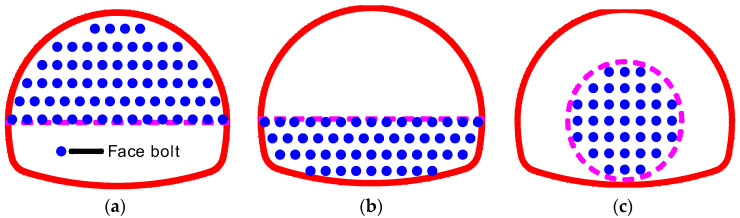
Different reinforcement range. (**a**) Upper reinforcement. (**b**) Lower reinforcement. (**c**) Central round reinforcement.

**Figure 18 materials-16-02193-f018:**
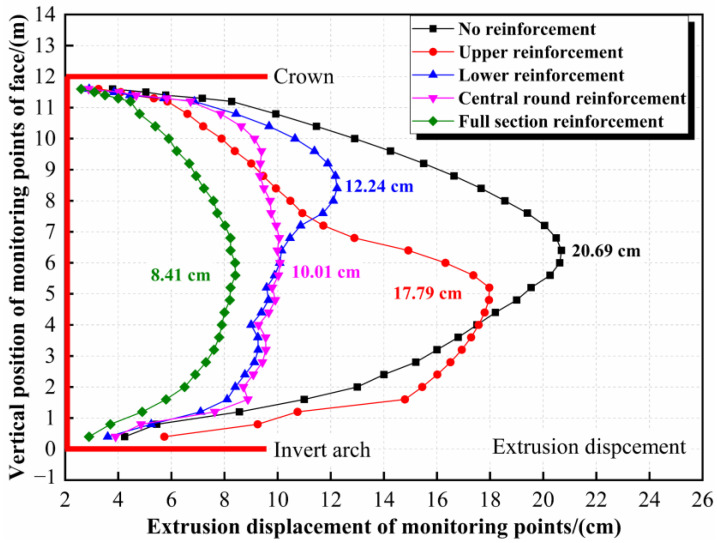
Extrusion displacement of target face under different reinforcement ranges.

**Table 1 materials-16-02193-t001:** GFRP pull-out test conditions.

Anchorage Length (m)	Mortar Strength (MPa)	Number of Specimens			Total
		Φ22	Φ25	Φ28	
0.5	M15	3	3	3	9
	M20	3	3	3	9
	M25	3	3	3	9
1.0	M20	3	3	3	9
1.5		3	3	3	9
2.0		3	3	3	9
Total		18	18	18	54

**Table 2 materials-16-02193-t002:** Mechanical parameters of GFRP in different diameters.

Diameter (mm)	Effective Diameter (mm)	Tensile Strength (MPa)	Young’s Modulus (GPa)	Design Value of Tensile Strength (MPa)
22	21.8	634.2	45.3	443.4
25	24.8	671.2	43.3	469.8
28	27.7	692.1	41.2	484.5

**Table 3 materials-16-02193-t003:** Mortar mix proportions in different strength grades.

Mortar Strength (MPa)	Cement (kg)	Sand (kg)	Water-Cement Ratio	Water (kg)
15	310	1500	0.84	260
20	710	1219	0.38	271
25	898	898	0.45	404

**Table 4 materials-16-02193-t004:** Results of Bond and Pull-out Test on GFRP Bolts.

Anchorage Length (m)	Mortar Strength (MPa)	GFRP Bars’ Diameter (mm)	Mean Ultimate Pull-Out Resistance (kN)	Mean Bond Strength (MPa)
0.5	M15	22	111.02	3.21
		25	109.91	2.80
		28	118.33	2.69
	M20	22	179.26	5.19
		25	186.71	4.75
		28	207.30	4.71
	M25	22	217.04	6.28
		25	234.47	5.97
		28	235.32	5.35
1.0	M20	22	266.14	3.85
		25	277.08	3.53
		28	282.81	3.22
1.5		22	289.45	2.79
		25	373.82	3.17
		28	391.68	2.97
2.0		22	265.45	1.92
		25	402.34	2.56
		28	459.12	2.61

**Table 5 materials-16-02193-t005:** Calculated values of grout cohesion under different conditions.

Working Conditions	Cohesive Strength of Grouting/(kN/m)		cg2cg1
	$c_{g1}$	$c_{g2}$	
M15–22	225.32	518.36	2.3
M15–25	190.07	589.05	3.1
M15–28	184.73	659.73	3.6
M20–22	342.12	691.15	2.0
M20–25	357.36	785.40	2.2
M20–28	371.21	879.65	2.4
M25–22	468.60	863.94	1.8
M25–25	487.73	981.75	2.0
M25–28	436.30	1099.56	2.5

**Table 6 materials-16-02193-t006:** Grout parameters for mortar specimen M20.

Working Conditions	$c_{g}$				$k_{g}$
	0.25	0.50	0.75	1.00	
M20–22	85.53	171.06	256.59	342.12	776
M20–25	89.34	178.68	268.02	357.36	889
M20–28	92.80	185.61	278.41	371.21	1020

**Table 7 materials-16-02193-t007:** Parameters of surrounding rock and initial support.

	Young’s Modulus *E* (GPa)	Poisson Ration *v*	Volumetric Weight γ (kN/m^3^)	Cohesion *C* (MPa)	Internal Friction Angle φ(°)
Surrounding rock	0.02	0.40	18.0	0.04	8.0
Initial support	20	0.25	22.0	—	—

**Table 8 materials-16-02193-t008:** Calculation parameters of GFPR face bolts.

Effective Diameter of Reinforcement *d* (mm)	Reinforcement Length *l* (m)	Design Value of Tension Strength *σ*_adm_ (MPa)	Bond Strength *τ*_adm_ (MPa)	Tunnel Face Area *A*s (m^2^)
24.8	17	469.8	2.56	149.23

**Table 9 materials-16-02193-t009:** Maximum displacement and reinforcement effectiveness of tunnel face.

Number of Face Bolts	0	77	90	112	150	187
Maximum extrusion displacement	20.7	11.2	10.5	10.4	8.4	7.6
Maximum displacement reduction rate	—	45.9%	47.8%	49.8%	59.4%	63.3%

## Data Availability

Not applicable.

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
