# Peer review of "Mechanical Properties of GFRP Bolts and Its Application in Tunnel Face Reinforcement"

_materials, 2023, doi:10.3390/ma16062193_

Round 1

Reviewer 1 Report

The article "Mechanical Properties of GFRP Bolts and Its Application in Tunnel Face Reinforcement" is interesting. However, a few minor comments are given below. 

1) "Corresponding author. E-mail address: 18190201538@163.com" is this the correct corresponding email?

2) Abstract needs revision with some quantitative results.

3) Section 2 should be renamed as the methodology 

4) Line 120, "For manufacturing standards" Provide necessary references. 

5) Figure no 1 is not quality enough for a research paper. The author must take proper sample pictures and label them to identify clearly in the figure. 

6) section "2.4. Analysis of test results" can be a separate main section. 

7)  Author must provide more explanation about figure 3.

8) For readers to quickly catch the contribution in this work, it would be better to highlight major difficulties and challenges, and authors' original achievements to overcome them, in a clearer way in Introduction section. Also, the recent Literature available in this field should be included in the introduction section.

9) Moreover, the results and discussion are not clearly dealt the outcomes of the proposed work. The authors should explicitly state the novel contribution of this work, the similarities, and the differences of this work with the previous publications in this section.

Author Response

Response to Reviewer 1 Comments

Thank you very much for the reviewer’s helpful comments. The reviewer’s comments and suggestions are very helpful for us to improve the quality of the manuscript. Incorporating the reviewer’s kind and helpful points, the authors have revised the manuscript carefully and the responses are listed as follows.

Point 1: "Corresponding author. E-mail address: 18190201538@163.com" is this the correct corresponding email?

Response 1: Thanks to the Reviewer for noticing this point. This is the correct corresponding email.

Point 2: Abstract needs revision with some quantitative results.

Response 2: Thanks for your comments and suggestions. The authors agree with the Reviewer that some quantitative results in the Abstract could provide more scientific guidance to the readers. To reflect this, the part of abstract has been modified as follows. Firstly, the effects of rebar diameter, anchorage length and mortar strength on the bonding properties of GFRP bars were studied by indoor pull-out tests. The bond strength-slip curves under different working conditions were obtained, and the curves showed that the ultimate bond strength between GFRP bars and mortar was negatively correlated with the diameter of GFRP bars, but positively correlated with the strength of mortar. In addition, the increase of anchorage length would lead to the reduction of bonding strength. Secondly, the inverse analysis was used to analyse the mechanical parameters of bond performance of the anchor bars by finite difference software FLAC3D, and the results indicated that 1/5 of compressive strength of GFRP bar grouting body can be taken as the ultimate bond strength to calculate the cohesive strength of grout. And the formula of GFRP bar grouting body stiffness has been revised. Finally, based on the results of laboratory test and inverse analysis, the numerical simulation analysis results showed that the optimal reinforcement configuration for shallow buried tunnel face surrounded by the weak rock is to use GFRP bars with a length of 17 m to arrange in the center circle of the tunnel face with a reasonable reinforcement density of 1.0 bolt/m2. The calculation formula of stiffness and cohesion strength of GFRP bar grouting body and the reinforcement scheme proposed in this paper can provide reference for the construction of shallow buried soft surrounding rock tunnel.

The above modifications have been retained in the original text.

Point 3: Section 2 should be renamed as the methodology.

Response 3: Thanks for your comments and suggestions. The authors have revised the title of Section 2as ‘Experimental program’.

The above modifications have been retained in the original text.

Point 4: Line 120, "For manufacturing standards" Provide necessary references.

Response 4: Thanks for your comments and suggestions. The authors have inserted the reference for the manufacturing standard in the body text.

'Standards for Test Method for Concrete Structures' [1] is an industry standard published by the Ministry of Housing and Urban-Rural Development of the People 's Republic of China, and the production of concrete samples is carried out with reference to the specification in this paper. In order to make readers understand this passage better, the passage has been modified in the text as: The shape of the specimen is cuboid, and the four sizes are 0.3mx0.3mx0.5m, 0.3mx0.3mx1m, 0.3mx0.3mx1.5m and 0.3mx0.3mx2m respectively. Three specimens are made for each size, and the appearance of the specimens is shown in Fig.1. And the standard reference of specimen making is 'Standards for Test Method for Concrete Structures’. In addition, the steel casing is set at the free end of the bar in order to avoid the bar being clipped during the loading process.

The above modifications have been retained in the original text.

Point 5: Figure no 1 is not quality enough for a research paper. The author must take proper sample pictures and label them to identify clearly in the figure.

Response 5: Thanks forcatching this. The authors would like to apologize for the possible confusion caused by the low image quality, arising from the limitations of test equipment. The authors have improved the Fig.1 by adding some sketch illustration, as shown in the following:

Fig. 1 Specimen appearance

The above modifications have been retained in the original text.

Point 6: section "2.4. Analysis of test results" can be a separate main section.

Response 6: As suggested by the Reviewer, Section '2.4.Analysis of test results' has been updated as the Section '3.Experimental results and Discussion'

The above modifications have been retained in the original text.

Point 7: Author must provide more explanation about figure 3.

Response 7: Thanks for your comments and suggestions. To reflect this, the authors have provided more interpretations regarding the content invovled in Fig.3. Corresponding revisions are as follows: The ultimate bond strength of bolts with different diameters is shown in Fig.3. For example, the bond strength of GFRP bolt with diameter of 22 mm, 25 mm and 28 mm is 3.21 MPa, 2.80 MPa and 2.69 MPa respectively when the mortar strength is M15, the bond strength of GFRP bolt with diameter of 22 mm, 25 mm and 28 mm is 5.19 MPa, 4.75 MPa and 4.71 MPa respectively when the mortar strength is M20, and when the mortar strength is M28, the bond strength of GFRP bolt with diameter of 22 mm, 25 mm and 28 mm is 6.28 MPa, 5.97 MPa and 5.35 MPa respectively. It can be seen from the results that with the increase of bolt diameter, the ultimate bond strength decreases.

The above modifications have been retained in the original text.

Point 8: For readers to quickly catch the contribution in this work, it would be better to highlight major difficulties and challenges, and authors' original achievements to overcome them, in a clearer way in Introduction section. Also, the recent Literature available in this field should be included in the introduction section.

Response 8: Thanks for your comments and suggestions. We further highlight the major difficulties and challenges in the introduction and improved the writing as: The effects of rebar diameter, anchorage length and mortar strength on the bonding properties of GFRP bars were studied based on the pull-out tests of GFRP bars. Then, the bond pull-out tests of GFRP bars were simulated by the finite difference software FLAC3D. An inverse analysis was subsequently performed in order to reasonbly quantify the mechanical parameters of GFPR bars-mortar grouting body. Finally, the effects of length, reinforcement density and range of GFRP reinforcement on the face of shallow buried tunnel in soft stratum was studied by FLAC3D.

In order to further emphasize the importance of this study, we have refferred to the following works published most recently in the introduction:

Xu et al. [2] proposed the scheme of adding prefabricated corrugated steel plate to the supporting structure. Prefabricated corrugated steel plate can effectively ensure the stability of the tunnel face and surrounding rock, reduce its convergence deformation, and ensure the safety of subway tunnel construction through numerical simulation and field verification.

Wang et al. [3] proposed a 3D rotational silo-torus stability model for face analysis of circular tunnel faces constructed under cohesive-frictional soils. Centrifugal tests and numerical simulations show that the model can accurately describe the failure mode of roadway working face.

Nguyen et al. [4] indicated that the bond strength between fiber and concrete matrix depends not only on the properties of the slurry, but also on the shape and surface of the fiber.

Zhang et al. [5] found that the safety factor of tunnel face is positively correlated with friction angle, cohesion, bolt length and number of bolts. However, if the bolt length and the bolt density exceed the optimal values, the safety will not be greatly improved, but will increase the construction cost.

Zhang et al. [6] proposed that bolt support can reduce the support pressure of the tunnel face and the stress release degree of the surrounding rock, and reduce the scope of the failure zone.

Shakiba et al. [7] found that the strength of concrete affects the pull-out failure mode of the anchorage specimen through drawing pull-out tests out on the sample, and The failure of high-strength concrete is due to the shedding of the surface of GFRP bars, while the failure of low-strength concrete is due to the crushing of concrete.

Kuang et al. [8] found that the ultimate bearing capacity and the average bond strength of the first interface of GFRP anchor bar are 25 kN and 9.4% higher than those of steel anchor, respectively, but the average bond strength of the second interface of steel anchor is slightly higher than that of GFRP anchor bar.

Kim et al. [9] found that the appropriate water-to-cement ratio can improve the bond strength between GFRP bars and concrete. When the water-to-cement ratio is constant, the higher the concrete strength, the higher the bond strength between concrete and GFRP bars.

The above modifications have been retained in the original text.

Point 9: Moreover, the results and discussion are not clearly dealt the outcomes of the proposed work. The authors should explicitly state the novel contribution of this work, the similarities, and the differences of this work with the previous publications in this section.

Response 9: Thanks for your comments and suggestions. The section of conclusions have been refined to highlight the novel contributions of this paper. The revised conclusions are: The effects of rebar diameter, anchorage length and mortar strength on the bonding properties of GFRP bars were studied based on the indoor bond pull-out tests of GFRP bars. Then, the bond pull-out tests of GFRP bars were simulated by finite difference software FLAC3D and the inverse analysis of related anchorage parameters in numerical calculation is carried out. We also present formulas for calculating the stiffness and cohesive strength of GFRP bar grouting body. Based to the results of inversion analysis, the optimal reinforcement length, the reasonable reinforcement density, and the reasonable reinforcement range of the GFRP bar anchors was analycally determined in the shallow soft surrounding rock. The main conclusions include:

(1) Laboratory test results show that the ultimate bond strength between GFRP bolts and mortar is negatively correlated with bolt diameter, but positively correlated with mortar strength when the anchorage length is constant. Additionly, when the bolt diameter and mortar strength are constant, the bond strength is reduced with the increase of the anchorage length.

(2) Split failures occur in all M15 mortar specimens when the anchorage length is 0.5 m. As a result, that grade mortar shall be avoided as the grouting body during the real construction. In addition, by analyzing the proportion of failure modes of specimens, it is concluded that in engineering practice, the strength of mortar used as grouting material shall be greater than M20. When M20 mortar are used as grouting material, the minimum anchorage length of GFRP bolts with diameter of 22 mm, 25 mm, 28 mm are 1.0 m, 1.5 m, 2.0 m, respectively.

(3) Based on the ultimate bond strength of GFRP bars-mortar obtained from laboratory tests, the mechanical parameters of GFRP bars-mortar grouting body is obtained by an inverse analysis. Among them, the cohesive strength of the grouting body can be calculated by 1/5 of the compressive strength of the grouting body as the ultimate bond strength. The stiffness of grouting body can be obtained by multiplying the stiffness formula of grouting body of reinforced bolt-concrete pull-out model with reduction factor, and the range of reduction factor is .

(4)A failure model of shallow tunnel face in soft stratum is developed to determine the reasonable reinforcement length by simplifying the tunnel face failure model proposed by Davis. Considering the construction safety and cost, the optimum reinforcement length is taken as 17 m, the reasonable reinforcement density is 1.0 bolt/m2, and the GFRP bolts are anchored in the range of central round.

The above modifications have been retained in the original text.

Thank you so much for your valuable comments on this article, it will be a treasure for the rest of my life. Happy life to you!

References

  1. Ministry of Housing and Urban-Rural Development of the People’s Republic of China GB/T 50152-2012 Standard for test method of concrete structures; China Architecture & Building Press: Beijing, 2012;
  2. Xu, P.; Wei, Y.; Yang, Y.; Zhou, X. Application of Fabricated Corrugated Steel Plate in Subway Tunnel Supporting Structure. Case Studies in Construction Materials 2022, 17, e01323, doi:10.1016/j.cscm.2022.e01323.
  3. Wang, C.; Hou, J.; Chen, Y.-M.; Ye, X.-W.; Chu, W.-J. A 3D Rotational Silo-Torus Model for Face Stability Analysis of Circular Tunnels. Case Studies in Construction Materials 2023, 18, e01736, doi:10.1016/j.cscm.2022.e01736.
  4. Nguyen, N.T.; Bui, T.-T.; Bui, Q.-B. Fiber Reinforced Concrete for Slabs without Steel Rebar Reinforcement: Assessing the Feasibility for 3D-Printed Individual Houses. Case Studies in Construction Materials 2022, 16, e00950, doi:10.1016/j.cscm.2022.e00950.
  5. Zhang, X.; Wang, M.; Wang, Z.; Li, J.; Tong, J.; Liu, D. A Limit Equilibrium Model for the Reinforced Face Stability Analysis of a Shallow Tunnel in Cohesive-Frictional Soils. Tunnelling and Underground Space Technology 2020, 105, 103562, doi:10.1016/j.tust.2020.103562.
  6. Zhang, X.; Wang, M.; Lyu, C.; Tong, J.; Yu, L.; Liu, D. Experimental and Numerical Study on Tunnel Faces Reinforced by Horizontal Bolts in Sandy Ground. Tunnelling and Underground Space Technology 2022, 123, 104412, doi:10.1016/j.tust.2022.104412.

    Shakiba, M.; Hosseini, S.M.; Bazli, M.; Mortazavi, S.M.R.; Ghobeishavi, M.A. Enhancement of the Bond Behaviour between Sand Coated GFRP Bar and Normal Concrete Using Innovative Composite Anchor Heads. Mater Struct 2022, 55, 236, doi:10.1617/s11527-022-02074-9.

    1. Kuang, Z.; Zhang, M.; Bai, X. Load-Bearing Characteristics of Fibreglass Uplift Anchors in Weathered Rock. Proceedings of the Institution of Civil Engineers - Geotechnical Engineering 2020, 173, 49–57, doi:10.1680/jgeen.18.00195.
    2. Kim, J.; Jeong, S.; Kim, H.; Kim, Y.; Park, S. Bond Strength Properties of GFRP and CFRP According to Concrete Strength. Applied Sciences 2022, 12, 10611, doi:10.3390/app122010611.

Reviewer 2 Report

The paper "Mechanical Properties of GFRP Bolts and Its Application in Tunnel Face Reinforcement" presents a relevant theme and within the scope of this journal, and can be considered after some corrections suggested below:

(a) The abstract is generally well written, however in terms of content it is generic, i.e., the authors lack an in-depth study of the quantitative results of this research;

(b) Scientific innovation is limited in the introduction of the paper, the authors must go deeper and detail what this research differs from countless others that exist on this topic, this must be evidenced together with the objectives at the end of the introduction;

(c) The state of the art of the evaluated topic needs to be improved by the authors, note that some topics are absent and need to be known with current research, such as: 10.1016/j.cscm.2022.e00950; 10.1016/j.cscm.2022.e01323; 10.1016/j.cscm.2022.e01736.

(d) The paper, in general, is outside the Materials format, in addition, in some composite figures the authors do not explain what each one refers to in their respective legend;

(e) “A fixed speed v was put axially along the reinforcement bolts at their free ends to simulate the 186 pulling out until the failure of the bolt grouting. Several monitoring points were set up in the interface 187 between the bolts and the specimens to note down the pull-out displacement, and the fish function was 188 written to record the bond strength of the bolts, thus obtaining the curve of bond strength-slippage based 189 on the numerical simulation test” Authors should better explain this excerpt from the paper;

(f) “In view of that there is no relevant specification to explain the reinforcement parameters of face 284 bolts in the tunnel engineering, and considering the similarity between the tunnel face support and the 285 building foundation pit support, the reinforcement length of tunnel face bolts into stable surrounding 286 rock is taken as 1.5 m based on the Technical Specification for Retaining and Protection of Building 287 Foundations Excavations [30], therefore the minimum theoretical reinforcement length in tunnel vault is 288 taken as:”  Authors should better explain this excerpt from the paper;

Author Response

Response to Reviewer 2 Comments

Thank you very much for the reviewer’s helpful comments. The reviewer’s comments and suggestions are very helpful for us to improve the quality of the manuscript. Incorporating the reviewer’s kind and helpful points, the authors have revised the manuscript carefully and the responses are listed as follows.

Point 1: The abstract is generally well written, however in terms of content it is generic, i.e., the authors lack an in-depth study of the quantitative results of this research.

Response 1: Thanks for your comments and suggestions. The authors agree with the Reviewer that some quantitative results in the Abstract could provide more scientific guidance to the readers. To reflect this, the part of abstract has been modified as follows. Firstly, the effects of rebar diameter, anchorage length and mortar strength on the bonding properties of GFRP bars were studied by indoor pull-out tests. The bond strength-slip curves under different working conditions were obtained, and the curves showed that the ultimate bond strength between GFRP bars and mortar was negatively correlated with the diameter of GFRP bars, but positively correlated with the strength of mortar. In addition, the increase of anchorage length would lead to the reduction of bonding strength. Secondly, the inverse analysis was used to analyse the mechanical parameters of bond performance of the anchor bars by finite difference software FLAC3D, and the results indicated that 1/5 of compressive strength of GFRP bar grouting body can be taken as the ultimate bond strength to calculate the cohesive strength of grout. And the formula of GFRP bar grouting body stiffness has been revised. Finally, based on the results of laboratory test and inverse analysis, the numerical simulation analysis results showed that the optimal reinforcement configuration for shallow buried tunnel face surrounded by the weak rock is to use GFRP bars with a length of 17 m to arrange in the center circle of the tunnel face with a reasonable reinforcement density of 1.0 bolt/m2. The calculation formula of stiffness and cohesion strength of GFRP bar grouting body and the reinforcement scheme proposed in this paper can provide reference for the construction of shallow buried soft surrounding rock tunnel.

The above modifications have been retained in the original text.

Point 2: Scientific innovation is limited in the introduction of the paper, the authors must go deeper and detail what this research differs from countless others that exist on this topic, this must be evidenced together with the objectives at the end of the introduction.

Response 2: Thanks for your comments and suggestions. We further highlight the major difficulties and challenges in the introduction and improved the writing as: The effects of rebar diameter, anchorage length and mortar strength on the bonding properties of GFRP bars were studied based on the  pull-out tests of GFRP bars. Then, the bond pull-out tests of GFRP bars were simulated by the finite difference software FLAC3D. An inverse analysis was subsequently performed in order to reasonbly quantify the mechanical parameters of GFPR bars-mortar grouting body. Finally, the effects of length, reinforcement density and range of GFRP reinforcement on the face of shallow buried tunnel in soft stratum was studied by FLAC3D.

Meanwhile, Some of the conclusions have been modified to highlight the recent contributions of this paper. The revised conclusions are: The effects of rebar diameter, anchorage length and mortar strength on the bonding properties of GFRP bars were studied based on the indoor bond pull-out tests of GFRP bars. Then, the bond pull-out tests of GFRP bars were simulated by finite difference software FLAC3D and the inverse analysis of related anchorage parameters in numerical calculation is carried out. We also present formulas for calculating the stiffness and cohesive strength of GFRP bar grouting body. Based to the results of inversion analysis, the optimal reinforcement length, the reasonable reinforcement density, and the reasonable reinforcement range of the GFRP bar anchors was analycally determined in the shallow soft surrounding rock.The main conclusions include:

(1) Laboratory test results show that the ultimate bond strength between GFRP bolts and mortar is negatively correlated with bolt diameter, but positively correlated with mortar strength when the anchorage length is constant. Additionly, when the bolt diameter and mortar strength are constant, the bond strength is reduced with the increase of the anchorage length.

(2) Split failures occur in all M15 mortar specimens when the anchorage length is 0.5 m. As a result, that grade mortar shall be avoided as the grouting body during the real construction. In addition, by analyzing the proportion of failure modes of specimens, it is concluded that in engineering practice, the strength of mortar used as grouting material shall be greater than M20. When M20 mortar are used as grouting material, the minimum anchorage length of GFRP bolts with diameter of 22 mm, 25 mm, 28 mm are 1.0 m, 1.5 m, 2.0 m, respectively.

(3) Based on the ultimate bond strength of GFRP bars-mortar obtained from laboratory tests, the mechanical parameters of GFRP bars-mortar grouting body is obtained by an inverse analysis. Among them, the cohesive strength of the grouting body can be calculated by 1/5 of the compressive strength of the grouting body as the ultimate bond strength. The stiffness of grouting body can be obtained by multiplying the stiffness formula of grouting body of reinforced bolt-concrete pull-out model with reduction factor, and the range of reduction factor is .

(4)A failure model of shallow tunnel face in soft stratum is developed to determine the reasonable reinforcement length by simplifying the tunnel face failure model proposed by Davis. Considering the construction safety and cost, the optimum reinforcement length is taken as 17 m, the reasonable reinforcement density is 1.0 bolt/m2, and the GFRP bolts are anchored in the range of central round.

The above modifications have been retained in the original text.

Point 3: The state of the art of the evaluated topic needs to be improved by the authors, note that some topics are absent and need to be known with current research, such as: 10.1016/j.cscm.2022.e00950; 10.1016/j.cscm.2022.e01323; 10.1016/j.cscm.2022.e01736.

Response 3: Thanks for your comments and suggestions. In order to further emphasize the importance of this study, we quote the following latest research literature in the introduction.

Xu et al. [1] proposed the scheme of adding prefabricated corrugated steel plate to the supporting structure. Prefabricated corrugated steel plate can effectively ensure the stability of the tunnel face and surrounding rock, reduce its convergence deformation, and ensure the safety of subway tunnel construction through numerical simulation and field verification.

Wang et al. [2] proposed a 3D rotational silo-torus stability model for face analysis of circular tunnel faces constructed under cohesive-frictional soils. Centrifugal tests and numerical simulations show that the model can accurately describe the failure mode of roadway working face.

Nguyen et al. [3] indicated that the bond strength between fiber and concrete matrix depends not only on the properties of the slurry, but also on the shape and surface of the fiber.

Zhang et al. [4] found that the safety factor of tunnel face is positively correlated with friction angle, cohesion, bolt length and number of bolts. However, if the bolt length and the bolt density exceed the optimal values, the safety will not be greatly improved, but will increase the construction cost.

Zhang et al. [5] proposed that bolt support can reduce the support pressure of the tunnel face and the stress release degree of the surrounding rock, and reduce the scope of the failure zone.

Shakiba et al. [6] found that the strength of concrete affects the pull-out failure mode of the anchorage specimen through drawing pull-out tests out on the sample, and The failure of high-strength concrete is due to the shedding of the surface of GFRP bars, while the failure of low-strength concrete is due to the crushing of concrete.

Kuang et al. [7] found that the ultimate bearing capacity and the average bond strength of the first interface of GFRP anchor bar are 25 kN and 9.4% higher than those of steel anchor, respectively, but the average bond strength of the second interface of steel anchor is slightly higher than that of GFRP anchor bar.

Kim et al. [8] found that the appropriate water-to-cement ratio can improve the bond strength between GFRP bars and concrete. When the water-to-cement ratio is constant, the higher the concrete strength, the higher the bond strength between concrete and GFRP bars.

The above modifications have been retained in the original text.

Point 4: The paper, in general, is outside the Materials format, in addition, in some composite figures the authors do not explain what each one refers to in their respective legend.

Response 4: Thanks for your comments and suggestions. To better illustrate the pictures,, we have revised the legend of Figure 1, Figure 4, Figure 9, and Figure 11 in the text.

Fig. 1 Specimen appearance

Fig. 4 Relationship between bond strength and the mortar strength

Fig. 9 Relationship between computed ultimate bond strength and grout cohesion

Fig. 11 Numerical analysis of model

The above modifications have been retained in the original text.

Point 5: “A fixed speed v was put axially along the reinforcement bolts at their free ends to simulate the 186 pulling out until the failure of the bolt grouting. Several monitoring points were set up in the interface 187 between the bolts and the specimens to note down the pull-out displacement, and the fish function was 188 written to record the bond strength of the bolts, thus obtaining the curve of bond strength-slippage based 189 on the numerical simulation test” Authors should better explain this excerpt from the paper.

Response 5: Thanks for your comments and suggestions. We are so sorry for the confusion here. The revelant sentence has been revised as: As shown in Figure 8, one end of the GFRP bar is pulled out at speed V until the connection between the bolt and the mortar specimen is destroyed [9]. Monitoring points are set on the contact surface between GFRP bars and mortar specimens to record the displacement changes of anchors during the drawing process. Finally, the bond strength of the bolt is calculated by writing the fish function to obtain the bond strength versus slip displacement curve.

The above modifications have been retained in the original text.

Point 6: “In view of that there is no relevant specification to explain the reinforcement parameters of face 284 bolts in the tunnel engineering, and considering the similarity between the tunnel face support and the 285 building foundation pit support, the reinforcement length of tunnel face bolts into stable surrounding 286 rock is taken as 1.5 m based on the Technical Specification for Retaining and Protection of Building 287 Foundations Excavations [30], therefore the minimum theoretical reinforcement length in tunnel vault is 288 taken as:”  Authors should better explain this excerpt from the paper.

Response 6: Thanks for your comments and suggestions. ‘Technical Specification for Retaining and Protection of Building Foundation Excavations’ [10] is an industry standard formulated by the Ministry of Housing and Urban-Rural Development of the People's Republic of China. Article 4.7.9 stipulates that the anchorage length of anchor bolt should not be less than 1.5 m. Therefore, we have modified this part of the original text as follows: It should be noted that no relevant specification to explain the length parameters of the bolt-reinforced tunnel face in the tunnel engineering. Considering the similarity between the tunnel face support and the foundation pit support, the anchorage length of the bolt is determined to be not less than 1.5m according to the 'Technical Specification for Retaining and Protection of Building Foundation Excavations'.

The above modifications have been retained in the original text.

Thank you so much for your valuable comments on this article, it will be a treasure for the rest of my life. Happy life to you!

References

  1. Xu, P.; Wei, Y.; Yang, Y.; Zhou, X. Application of Fabricated Corrugated Steel Plate in Subway Tunnel Supporting Structure. Case Studies in Construction Materials 2022, 17, e01323, doi:10.1016/j.cscm.2022.e01323.
  2. Wang, C.; Hou, J.; Chen, Y.-M.; Ye, X.-W.; Chu, W.-J. A 3D Rotational Silo-Torus Model for Face Stability Analysis of Circular Tunnels. Case Studies in Construction Materials 2023, 18, e01736, doi:10.1016/j.cscm.2022.e01736.
  3. Nguyen, N.T.; Bui, T.-T.; Bui, Q.-B. Fiber Reinforced Concrete for Slabs without Steel Rebar Reinforcement: Assessing the Feasibility for 3D-Printed Individual Houses. Case Studies in Construction Materials 2022, 16, e00950, doi:10.1016/j.cscm.2022.e00950.
  4. Zhang, X.; Wang, M.; Wang, Z.; Li, J.; Tong, J.; Liu, D. A Limit Equilibrium Model for the Reinforced Face Stability Analysis of a Shallow Tunnel in Cohesive-Frictional Soils. Tunnelling and Underground Space Technology 2020, 105, 103562, doi:10.1016/j.tust.2020.103562.
  5. Zhang, X.; Wang, M.; Lyu, C.; Tong, J.; Yu, L.; Liu, D. Experimental and Numerical Study on Tunnel Faces Reinforced by Horizontal Bolts in Sandy Ground. Tunnelling and Underground Space Technology 2022, 123, 104412, doi:10.1016/j.tust.2022.104412.
  6. Shakiba, M.; Hosseini, S.M.; Bazli, M.; Mortazavi, S.M.R.; Ghobeishavi, M.A. Enhancement of the Bond Behaviour between Sand Coated GFRP Bar and Normal Concrete Using Innovative Composite Anchor Heads. Mater Struct 2022, 55, 236, doi:10.1617/s11527-022-02074-9.
  7. Kuang, Z.; Zhang, M.; Bai, X. Load-Bearing Characteristics of Fibreglass Uplift Anchors in Weathered Rock. Proceedings of the Institution of Civil Engineers - Geotechnical Engineering 2020, 173, 49–57, doi:10.1680/jgeen.18.00195.
  8. Kim, J.; Jeong, S.; Kim, H.; Kim, Y.; Park, S. Bond Strength Properties of GFRP and CFRP According to Concrete Strength. Applied Sciences 2022, 12, 10611, doi:10.3390/app122010611.
  9. Itasca Consulting Group Inc. FLAC3D Version 6.0 Users’ Manual; US, Minnesota, 2019;
  10. Ministry of Housing and Urban Rural Development of the People’s Republic of China JGJ 120-2012 Technical Specification for Retaining and Protection of Building Foundation Excavations; China Architecture & Building Press: Beijing, 2012;

Reviewer 3 Report

The article "Mechanical Properties of GFRP Bolts and Its Application in Tunnel Face Reinforcement" is interesting. But some comments are given below to improve the paper, 

1) Highlight the main Objectives and aims of the study in the abstract. The abstract needs revision with some quantitative results.

2) Some more latest studies are required in the introduction section to further highlight the importance of this study.

3) For readers to quickly catch the contribution of this work, it would be better to highlight major difficulties and challenges, and the authors' original achievements to overcome them, in a clearer way in the Introduction section.

4) Moreover, the results and discussion are not clearly dealt with the outcomes of the proposed work. The authors should explicitly state the novel contribution of this work, and the similarities, and differences between this work with the previous publications in this section.

Author Response

Thank you very much for the reviewer’s helpful comments. The reviewer’s comments and suggestions are very helpful for us to improve the quality of the manuscript. Incorporating the reviewer’s kind and helpful points, the authors have revised the manuscript carefully and the responses are listed as follows.

Point 1: Highlight the main Objectives and aims of the study in the abstract. The abstract needs revision with some quantitative results.

Response 1: Thanks for your comments and suggestions. The authors agree with the Reviewer that some quantitative results in the Abstract could provide more scientific guidance to the readers. To reflect this, the part of abstract has been modified as follows. Firstly, the effects of rebar diameter, anchorage length and mortar strength on the bonding properties of GFRP bars were studied by indoor pull-out tests. The bond strength-slip curves under different working conditions were obtained, and the curves showed that the ultimate bond strength between GFRP bars and mortar was negatively correlated with the diameter of GFRP bars, but positively correlated with the strength of mortar. In addition, the increase of anchorage length would lead to the reduction of bonding strength. Secondly, the inverse analysis was used to analyse the mechanical parameters of bond performance of the anchor bars by finite difference software FLAC3D, and the results indicated that 1/5 of compressive strength of GFRP bar grouting body can be taken as the ultimate bond strength to calculate the cohesive strength of grout. And the formula of GFRP bar grouting body stiffness has been revised. Finally, based on the results of laboratory test and inverse analysis, the numerical simulation analysis results showed that the optimal reinforcement configuration for shallow buried tunnel face surrounded by the weak rock is to use GFRP bars with a length of 17 m to arrange in the center circle of the tunnel face with a reasonable reinforcement density of 1.0 bolt/m2. The calculation formula of stiffness and cohesion strength of GFRP bar grouting body and the reinforcement scheme proposed in this paper can provide reference for the construction of shallow buried soft surrounding rock tunnel.

The above modifications have been retained in the original text.

Point 2: Some more latest studies are required in the introduction section to further highlight the importance of this study.

Response 2: Thanks for your comments and suggestions. In order to further emphasize the importance of this study, we quote the following latest research literature in the introduction.

Xu et al. [1] proposed the scheme of adding prefabricated corrugated steel plate to the supporting structure. Prefabricated corrugated steel plate can effectively ensure the stability of the tunnel face and surrounding rock, reduce its convergence deformation, and ensure the safety of subway tunnel construction through numerical simulation and field verification.

Wang et al. [2] proposed a 3D rotational silo-torus stability model for face analysis of circular tunnel faces constructed under cohesive-frictional soils. Centrifugal tests and numerical simulations show that the model can accurately describe the failure mode of roadway working face.

Nguyen et al. [3] indicated that the bond strength between fiber and concrete matrix depends not only on the properties of the slurry, but also on the shape and surface of the fiber.

Zhang et al. [4] found that the safety factor of tunnel face is positively correlated with friction angle, cohesion, bolt length and number of bolts. However, if the bolt length and the bolt density exceed the optimal values, the safety will not be greatly improved, but will increase the construction cost.

Zhang et al. [5] proposed that bolt support can reduce the support pressure of the tunnel face and the stress release degree of the surrounding rock, and reduce the scope of the failure zone.

Shakiba et al. [6] found that the strength of concrete affects the pull-out failure mode of the anchorage specimen through drawing pull-out tests out on the sample, and The failure of high-strength concrete is due to the shedding of the surface of GFRP bars, while the failure of low-strength concrete is due to the crushing of concrete.

Kuang et al. [7] found that the ultimate bearing capacity and the average bond strength of the first interface of GFRP anchor bar are 25 kN and 9.4% higher than those of steel anchor, respectively, but the average bond strength of the second interface of steel anchor is slightly higher than that of GFRP anchor bar.

Kim et al. [8] found that the appropriate water-to-cement ratio can improve the bond strength between GFRP bars and concrete. When the water-to-cement ratio is constant, the higher the concrete strength, the higher the bond strength between concrete and GFRP bars.

The above modifications have been retained in the original text.

Point 3: For readers to quickly catch the contribution of this work, it would be better to highlight major difficulties and challenges, and the authors' original achievements to overcome them, in a clearer way in the Introduction section.

Response 3: Thanks for your comments and suggestions. We further highlight the major difficulties and challenges in the introduction and improved the writing as: The effects of rebar diameter, anchorage length and mortar strength on the bonding properties of GFRP bars were studied based on the pull-out tests of GFRP bars. Then, the bond pull-out tests of GFRP bars were simulated by the finite difference software FLAC3D. An inverse analysis was subsequently performed in order to reasonbly quantify the mechanical parameters of GFPR bars-mortar grouting body. Finally, the effects of length, reinforcement density and range of GFRP reinforcement on the face of shallow buried tunnel in soft stratum was studied by FLAC3D.

The above modifications have been retained in the original text.

Point 4: Moreover, the results and discussion are not clearly dealt with the outcomes of the proposed work. The authors should explicitly state the novel contribution of this work, and the similarities, and differences between this work with the previous publications in this section.

Response 4: Thanks for your comments and suggestions. The section of conclusions have been refined to highlight the novel contributions of this paper. The revised conclusions are: The effects of rebar diameter, anchorage length and mortar strength on the bonding properties of GFRP bars were studied based on the indoor bond pull-out tests of GFRP bars. Then, the bond pull-out tests of GFRP bars were simulated by finite difference software FLAC3D and the inverse analysis of related anchorage parameters in numerical calculation is carried out. We also present formulas for calculating the stiffness and cohesive strength of GFRP bar grouting body. Based to the results of inversion analysis, the optimal reinforcement length, the reasonable reinforcement density, and the reasonable reinforcement range of the GFRP bar anchors was analycally determined in the shallow soft surrounding rock.The main conclusions include:

(1) Laboratory test results show that the ultimate bond strength between GFRP bolts and mortar is negatively correlated with bolt diameter, but positively correlated with mortar strength when the anchorage length is constant. Additionly, when the bolt diameter and mortar strength are constant, the bond strength is reduced with the increase of the anchorage length.

(2) Split failures occur in all M15 mortar specimens when the anchorage length is 0.5 m. As a result, that grade mortar shall be avoided as the grouting body during the real construction. In addition, by analyzing the proportion of failure modes of specimens, it is concluded that in engineering practice, the strength of mortar used as grouting material shall be greater than M20. When M20 mortar are used as grouting material, the minimum anchorage length of GFRP bolts with diameter of 22 mm, 25 mm, 28 mm are 1.0 m, 1.5 m, 2.0 m, respectively.

(3) Based on the ultimate bond strength of GFRP bars-mortar obtained from laboratory tests, the mechanical parameters of GFRP bars-mortar grouting body is obtained by an inverse analysis. Among them, the cohesive strength of the grouting body can be calculated by 1/5 of the compressive strength of the grouting body as the ultimate bond strength. The stiffness of grouting body can be obtained by multiplying the stiffness formula of grouting body of reinforced bolt-concrete pull-out model with reduction factor, and the range of reduction factor is .

(4)A failure model of shallow tunnel face in soft stratum is developed to determine the reasonable reinforcement length by simplifying the tunnel face failure model proposed by Davis. Considering the construction safety and cost, the optimum reinforcement length is taken as 17 m, the reasonable reinforcement density is 1.0 bolt/m2, and the GFRP bolts are anchored in the range of central round.

The above modifications have been retained in the original text.

Thank you so much for your valuable comments on this article, it will be a treasure for the rest of my life. Happy life to you!

Thank you so much for your valuable comments on this article, it will be a treasure for the rest of my life. Happy life to you!

References

  1. Xu, P.; Wei, Y.; Yang, Y.; Zhou, X. Application of Fabricated Corrugated Steel Plate in Subway Tunnel Supporting Structure. Case Studies in Construction Materials 2022, 17, e01323, doi:10.1016/j.cscm.2022.e01323.
  2. Wang, C.; Hou, J.; Chen, Y.-M.; Ye, X.-W.; Chu, W.-J. A 3D Rotational Silo-Torus Model for Face Stability Analysis of Circular Tunnels. Case Studies in Construction Materials 2023, 18, e01736, doi:10.1016/j.cscm.2022.e01736.
  3. Nguyen, N.T.; Bui, T.-T.; Bui, Q.-B. Fiber Reinforced Concrete for Slabs without Steel Rebar Reinforcement: Assessing the Feasibility for 3D-Printed Individual Houses. Case Studies in Construction Materials 2022, 16, e00950, doi:10.1016/j.cscm.2022.e00950.
  4. Zhang, X.; Wang, M.; Wang, Z.; Li, J.; Tong, J.; Liu, D. A Limit Equilibrium Model for the Reinforced Face Stability Analysis of a Shallow Tunnel in Cohesive-Frictional Soils. Tunnelling and Underground Space Technology 2020, 105, 103562, doi:10.1016/j.tust.2020.103562.
  5. Zhang, X.; Wang, M.; Lyu, C.; Tong, J.; Yu, L.; Liu, D. Experimental and Numerical Study on Tunnel Faces Reinforced by Horizontal Bolts in Sandy Ground. Tunnelling and Underground Space Technology 2022, 123, 104412, doi:10.1016/j.tust.2022.104412.
  6. Shakiba, M.; Hosseini, S.M.; Bazli, M.; Mortazavi, S.M.R.; Ghobeishavi, M.A. Enhancement of the Bond Behaviour between Sand Coated GFRP Bar and Normal Concrete Using Innovative Composite Anchor Heads. Mater Struct 2022, 55, 236, doi:10.1617/s11527-022-02074-9.
  7. Kuang, Z.; Zhang, M.; Bai, X. Load-Bearing Characteristics of Fibreglass Uplift Anchors in Weathered Rock. Proceedings of the Institution of Civil Engineers - Geotechnical Engineering 2020, 173, 49–57, doi:10.1680/jgeen.18.00195.
  8. Kim, J.; Jeong, S.; Kim, H.; Kim, Y.; Park, S. Bond Strength Properties of GFRP and CFRP According to Concrete Strength. Applied Sciences 2022, 12, 10611, doi:10.3390/app122010611.

Round 2

Reviewer 2 Report

ok.